# Learning Models for Actionable Recourse

**Alexis Ross**[*]
Harvard University
Allen Institute for Artificial Intelligence
alexisr@allenai.org

**Himabindu Lakkaraju**
Harvard University
hlakkaraju@seas.harvard.edu

**Osbert Bastani**
University of Pennsylvania
obastani@seas.upenn.edu

## Abstract

As machine learning models are increasingly deployed in high-stakes domains such as legal and financial decision-making, there has been growing interest in post-hoc methods for generating counterfactual explanations. Such explanations provide individuals adversely impacted by predicted outcomes (e.g., an applicant denied a loan) with *recourse*—i.e., a description of how they can change their features to obtain a positive outcome. We propose a novel algorithm that leverages adversarial training and PAC confidence sets to learn models that *theoretically guarantee* recourse to affected individuals with high probability. We demonstrate the efficacy of our approach with extensive experiments on real data.

## 1 Introduction

In recent years, there has been a growing interest in using machine learning to inform consequential decisions in legal and financial decision-making—e.g., deciding whether to give an applicant a loan [Hardt et al., 2016], bail to a defendant [Lakkaraju and Rudin, 2017], or parole to a prisoner [Zeng et al., 2017]. Because these decisions have an impact on the lives of the concerned individuals, it is critical to explain why the model made its prediction. Explanations are important not only to ensure that there are no issues with the way the prediction is made (e.g., making sure the decision is free of racial/gender bias [Hardt et al., 2016] and does not suffer from causal issues [Bastani et al., 2017]), but also to give the affected individual a justification for the decision. Thus, there has been a great deal of recent interest in explainable machine learning [Lou et al., 2012, Wang and Rudin, 2015, Ribeiro et al., 2016b, Lakkaraju et al., 2019].

We focus on counterfactual explanations [Wachter et al., 2017], which specify how features can be changed to obtain a different model prediction. These explanations can be used to provide individuals who are negatively impacted by model outcomes with *actionable recourses*—i.e., actions they can take to receive a positive outcome [Ustun et al., 2019]. For instance, for an applicant denied a loan, an actionable recourse might be: "to get a loan, increase your income by $1,000." Such a recourse must satisfy two properties to be actionable: (i) it only changes features that the individual can realistically modify—e.g., it cannot change race or gender, but can change income, and (ii) the magnitude of the change must be reasonable. An actionable recourse like this may be considered necessary in some settings because it provides an individual with agency over a consequential decision that affects them.

Prior research has addressed the need for recourses through post-hoc algorithms for computing individual recourses corresponding to certain kinds of models—e.g., using integer linear programming

---

[*]Work started at Harvard University.

35th Conference on Neural Information Processing Systems (NeurIPS 2021).

in the context of linear models [Ustun et al., 2019], or using gradient descent on the input for differentiable models [Wachter et al., 2017]. However, these approaches do not guarantee that actionable recourses exist; at best, Ustun et al. [2019] guarantee that they find one if it exists, and only for linear models. In other words, many affected individuals may not even be prescribed any actions that they can take to change their outcome.

We aim to provide tools for guaranteeing the existence of actionable recourses for domains in which recourses may be necessary. We propose a novel algorithm for learning models in a way that is designed to *ensure* the existence of actionable recourses so that affected individuals receive recourse with a high probability. To the best of our knowledge, our approach is the first to train models for which recourse is likely to exist with a high probability. It builds on adversarial training [Goodfellow et al., 2015a], which is designed to ensure that models are robust to adversarial examples. At a high level, given a binary classification model $f_\theta : \mathcal{X} \to \mathcal{Y}$, where $\mathcal{Y} = \{0, 1\}$, an adversarial example is a perturbation $\delta \in \Delta$ to an input $x \in \mathcal{X}$ such that $f_\theta(x + \delta) \neq f_\theta(x)$. In a typical adversarial training setting, we choose $\Delta$ to be "small" in some sense (e.g., in terms of $L_\infty$ norm of $\delta \in \Delta$) and aim to *guarantee* that adversarial examples do not exist—i.e., $f(x + \delta) = f(x)$ for all $\delta \in \Delta$.

In contrast, in our setting, we intuitively want to ensure that adversarial examples *do* exist. In particular, given an input $x$ for which $f(x) = 0$, we want there to exist recourse $\delta \in \Delta$ such that $f(x + \delta) = 1$, where in our case, $\Delta$ is the given set of all permissible recourses for the application domain. Thus, we adapt existing adversarial training algorithms to ensure the existence of recourse. As an added benefit, these algorithms can compute recourse significantly faster than existing techniques for general differentiable models [Wachter et al., 2017].

While adversarial training heuristically encourages recourse to exist, it does not provide any theoretical guarantees. We build on PAC confidence sets [Park et al., 2020] to guarantee that recourse exists with high probability (assuming the test distribution is the same as the training distribution).

We evaluate our approach on four real world datasets that cover lending, recidivism, bail, and credit outcomes. Our results demonstrate that our approach is very effective at improving recourse rates (i.e., the probability that individuals are given recourse) without noticeably reducing accuracy.

In addition, we show that we achieve this improvement in recourse rates without noticeably harming the quality of recourses or the brittleness of the underlying model. Firstly, we empirically demonstrate that our approach improves the rate at which models provide recourses that are *grounded in reality*: We find that our approach encourages the existence of recourses that both *obey causal constraints* driven by real-world causal relationships and are *in-distribution* to the original data. Secondly, we show that our approach encourages the existence of robust recourses (i.e., recourses that result in positive outcomes even when changed in small ways). Lastly, we show that these improvements in recourse rates do not render the underlying classifier brittle.[2]

**Related work.** Beyond Wachter et al. [2017], Ustun et al. [2019], other approaches have been proposed for generating recourses [Zhang et al., 2018, Hendricks et al., 2018, Mothilal et al., 2020, Looveren and Klaise, 2019, Poyiadzi et al., 2020, Karimi et al., 2020a,b,c]. However, all these works focus on how to *compute* recourse for given predictive models; in contrast, our goal is to *learn* predictive models that provide recourses at high rates. Any of these methods can be used in conjunction with ours. Our work also builds on adversarial training [Szegedy et al., 2014, Goodfellow et al., 2015b, Bastani et al., 2016, Shaham et al., 2018]. While recent work in model explainability has leveraged adversarial training to improve robustness of explanations [Lakkaraju et al., 2020b], their goal is to *reduce* the rate of adversarial examples, whereas ours is to *increase* the rate of recourses.

## 2 Problem Formulation

**Preliminaries.** Consider a binary classifier $f_\theta : \mathcal{X} \to \mathcal{Y}$, where $x \in \mathcal{X} \subseteq \mathbb{R}^{n_X}$ are the features, $y \in \mathcal{Y} = \{0, 1\}$ are the labels, and $\theta \in \Theta \subseteq \mathbb{R}^{n_\Theta}$ are the parameters. We assume that $f_\theta$ has the form $f_\theta(x) = \mathbb{1}(g_\theta(x) \geq \theta_0)$, where $g_\theta : \mathcal{X} \to [0, 1]$ is a scoring function and $\theta_0 \in \mathbb{R}$ is a decision threshold. We assume that $g_\theta$ is differentiable in $x$—i.e., $\nabla_x g_\theta(x)$ exists almost everywhere.[3]

---

[2]Our code is available at `https://github.com/alexisjihyeross/adversarial_recourse`.

[3]Note that we have focused on real-valued features. We discuss how our algorithm can be extended to handling categorical features in Section B.4.

**Recourse.** We seek to ensure that individuals given negative outcomes by $f_\theta$ are also given *recourse*.

**Definition 2.1.** Given a classifier $f_\theta : \mathcal{X} \to \mathcal{Y}$ and a set $\Delta \subseteq \mathbb{R}^{n_X}$, an input $x \in \mathcal{X}$ has *recourse* if there exists $\delta \in \Delta$ such that $f(x + \delta) = 1$.

We use $\mathcal{X}_\theta^R$ to denote the set of inputs for which recourse exists. The specific design of the set of permissible recourses $\Delta \subseteq \mathbb{R}^{n_X}$ is domain specific. We assume that $0 \in \Delta$; then, recourse automatically exists for positive outcomes $f_\theta(x) = 1$ by taking $\delta = 0$. In addition, we assume that $\Delta$ is a polytope—i.e., $\Delta = \{\delta \in \mathbb{R}^{n_X} \mid A\delta + b \geq 0\}$ for some $A \in \mathbb{R}^{k \cdot n_X}$ and $b \in \mathbb{R}^k$. That is, $\Delta$ can be expressed as a set of affine constraints. This assumption is required for computational tractability.

A standard choice is $\Delta = \{\delta \mid \|\delta\|_\infty \leq \delta_{\max}\}$, which says that the recourse can suggest changes to any feature by a bounded amount. We can apply this constraint after an affine transformation of $\delta$ that appropriately scales different covariates. In addition, we often want to restrict features—e.g., to avoid suggesting that an individual change their age, we can constrain $\delta_i = 0$, or to avoid suggesting that an individual decrease their income to qualify for a loan, we can constrain $\delta_i \geq 0$. In principle, $\Delta$ can also be tailored to individuals—e.g., disallow suggesting increased education for individuals who cannot afford to do so. Finally, $\Delta$ should be designed large enough so it includes a plausible recourse for every individual, yet small enough to ensure that the recourses do not overburden the individuals. All of these considerations are domain-specific; we describe our choices for datasets in our experiments in Section 4.

**Probably Approximately has REcourse (PARE).** Our goal is to ensure that recourse exists for most individuals. Given a confidence level $\epsilon \in \mathbb{R}_{>0}$, we say that $\theta$ *approximately has recourse* if

$$\mathbb{P}_{p(x)} \left[ x \in \mathcal{X}_\theta^R \right] \geq 1 - \epsilon,$$

i.e., recourse exists for $f_\theta$ with probability at least $1 - \epsilon$ w.r.t. the distribution $p(x)$ over individuals.

Then, our goal is to design an algorithm for estimating the model parameters $\theta$ so that $f_\theta$ approximately has recourse. To do so, our algorithm takes as input a held-out calibration dataset $Z \subseteq \mathcal{X} \times \mathcal{Y}$ of examples $(x, y) \sim p$, where $p(x, y)$ is the distribution over labeled examples, and outputs model parameters $\hat{\theta}(Z)$. Then, as in the probably approximately correct (PAC) learning framework [Valiant, 1984], our algorithm might additionally fail due to the randomness in $Z$. Thus, given a second confidence level $\alpha \in \mathbb{R}_{>0}$, we say that $\hat{\theta}$ *Probably Approximately has REcourse (PARE)* if

$$\mathbb{P}_{p(Z)} \left[ \hat{\theta}(Z) \text{ approximately has recourse} \right] \geq 1 - \alpha,$$

where $p(Z) = \prod_{(x,y) \in Z} p(x, y)$. In other words, our algorithm produces a model that approximately has recourse with probability at least $1 - \delta$ over $p(Z)$.

**Constructing a PARE classifier.** Note that we can trivially obtain a PARE classifier $f_\theta$ by choosing the decision threshold $\theta_0 = 0$, in which case $f_\theta(x) = 1$ for all $x \in \mathcal{X}$. However, this approach is undesirable since it assigns a positive outcome to all individuals. Instead, we want to maximize the performance of $f_\theta$ (e.g., in terms of accuracy, $F_1$ score, etc.) subject to a constraint that $f_\theta$ is PARE. Thus, we divide the problem of constructing a PARE classifier into two parts (i) increasing recourse rate: we train $g_\theta$ in a way that heuristically increases the rate at which inputs $x \in \mathcal{X}$ have recourse (for any $\theta_0$), and (ii) guaranteeing recourse: we choose $\theta_0$ to guarantee that the resulting $f_\theta$ is PARE.

## 3 Our Algorithm

We describe our algorithm for learning models that satisfy PARE while achieving good performance. We describe Step 1 (increasing recourse rate) in Section 3.1 and Step 2 (guaranteeing recourse) in Section 3.3. We describe ways to compute recourse in Section 3.2.

### 3.1 Step 1: Increasing Recourse Rate

**Background: adversarial training.** Consider a classifier $f_\theta : \mathcal{X} \to \mathcal{Y}$ and perturbations $\Delta \subseteq \mathbb{R}^{n_X}$. Given $x \in \mathcal{X}$, an *adversarial example* [Szegedy et al., 2014] for $x$ is a perturbation $\delta \in \Delta$ such that $f_\theta(x + \delta) \neq f_\theta(x)$—i.e., the perturbation $\delta$ (restricted to be small) changes the predicted label of $x$.

Adversarial examples are undesirable because they indicate that $f_\theta$ is not robust to small changes to the input that should not affect the class label (e.g., according to human predictions). Thus, there has

been a great deal of interest in designing algorithms for improving robustness to adversarial examples. The basic approach is *adversarial training* [Goodfellow et al., 2015b], which dynamically computes adversarial examples for inputs in the training set and adds these to the objective as additional training examples, as in data augmentation. In particular, given a loss function $\ell : \mathbb{R} \times \mathcal{Y} \to \mathbb{R}$, where $\ell(g_\theta(x), y)$ is the loss for training example $(x, y)$, they seek to compute

$$\theta^* = \underset{\theta \in \Theta}{\arg\min} \, \ell_A(\theta) \quad \text{where} \quad \ell_A(\theta) = \mathbb{E}_{p(x,y)} \left[ \ell(g_\theta(x), y) + \lambda \cdot \underset{\delta \in \Delta}{\max} \, \ell(g_\theta(x + \delta), y) \right].$$

In $\ell_A(\theta)$, the first term is the supervised learning loss, the second is the adversarially robust loss (i.e., encourage $g_\theta$ to be robust to adversarial examples $x + \delta$), and $\lambda \in \mathbb{R}_{\geq 0}$ is a hyperparameter.

The challenge in optimizing $\ell_A(\theta)$ is computing the maximum over $\delta \in \Delta$. To address this challenge, existing adversarial training algorithms leverage approximations enabling efficient computation of $\delta$.

**Our approach.** We use adversarial training to learn a model $f_\theta$ for which inputs $x \in \mathcal{X}$ have recourse at higher rates compared to models trained using conventional approaches. There are two key differences compared to adversarial training. First, we want recourse to exist, which corresponds to *encouraging* the existence of adversarial examples. Second, we only care about changing negative labels $f_\theta(x) = 0$ to positive ones $f_\theta(x + \delta) = 1$, not vice versa. Thus, we want to solve

$$\theta^* = \underset{\theta \in \Theta}{\arg\min} \, \ell_R(\theta) \quad \text{where} \quad \ell_R(\theta) = \mathbb{E}_{p(x,y)} \left[ \ell(g_\theta(x), y) + \lambda \cdot \underset{\delta \in \Delta}{\min} \, \ell(g_\theta(x + \delta), 1) \right]. \quad (1)$$

Compared to $\ell_A$, $\ell_R$ has a different second term in two ways: (i) the maximum over $\delta$ with a minimum, and (ii) the label $y$ is replaced with the label 1. We note that when $\lambda = 0$, Eq. 1 is supervised learning.

We optimize Eq. 1 using *adversarial training* [Goodfellow et al., 2015a, Shaham et al., 2018, Lakkaraju et al., 2020a], which performs stochastic gradient descent on $\ell_R(\theta)$. The key challenge to computing $\nabla_\theta \ell_R(\theta)$ is computing the gradient of the second term, which can be rewritten as follows:

$$\nabla_\theta \underset{\delta \in \Delta}{\min} \, \ell(g_\theta(x + \delta), 1) = \nabla_\theta \ell(g_\theta(x + \delta^*), 1),$$

where $\delta^*$ is the perturbation that maximizes the likelihood of positive outcome, or minimizes the loss between predicted and positive outcomes—i.e.,

$$\delta^* = \underset{\delta \in \Delta}{\arg\min} \, \ell(g_\theta(x + \delta), 1). \quad (2)$$

Computing $\delta^*$ is computationally challenging; thus, we use a Taylor approximation of the loss $\ell(g_\theta(x + \delta), 1) \approx \ell(g_\theta(x), 1) + \nabla_x \ell(g_\theta(x), 1)^\top \delta$. Using this approximation, Eq. 2 becomes

$$\delta^* = \underset{\delta \in \Delta}{\arg\min} \, \ell(g_\theta(x + \delta), 1) \approx \underset{\delta \in \Delta}{\arg\min} \, \nabla_x \ell(g_\theta(x), 1)^\top \delta,$$

where we have dropped the term $\ell(g_\theta(x), 1)$ since it is constant with respect to $\delta$. Finally, note that the optimization problem on the last step is a linear program (LP), since we have assumed that $\delta \in \Delta$ can be expressed as a set of affine constraints and since the objective is linear in $\delta$. In summary, we optimize Eq. 1 using stochastic gradient descent, where at each step we solve an LP to approximate the second term—i.e., given parameters $\theta_i$ and example $(x_i, y_i)$ on gradient step $i$, and step size $\eta_i \in \mathbb{R}_{>0}$ on step $i$, we use the stochastic gradient update

$$\theta_{i+1} = \theta_i + \eta_i \left( \nabla_\theta \ell(g_\theta(x), y) + \lambda \cdot \nabla_\theta \ell(g_\theta(x + \delta_i^*), 1) \right) \quad \text{where} \quad \delta_i^* = \underset{\delta \in \Delta}{\arg\min} \, \nabla_x \ell(g_\theta(x), 1)^\top \delta.$$

### 3.2 Computing Recourse

So far, we have focused on how to train a model $f_\theta$ that provides recourse. Once we have trained $f_\theta$, we still need a way to compute recourse for a given individual $x$—i.e., an algorithm $\mathcal{A} : \mathcal{X} \to \Delta$ for computing $\delta_x = \mathcal{A}(x)$ such that $f(x + \delta_x) = 1$. We describe three such algorithms; in general, any algorithm designed to output recourses can be used [Karimi et al., 2020a, Poyiadzi et al., 2020].

**Gradient descent.** The approach proposed in [Wachter et al.] [2017] can directly be applied to compute recourse. They solve the problem

$$\delta_x = \underset{\delta \in \Delta}{\arg\min} \{ \ell(g_\theta(x + \delta, 1) + \lambda' \cdot \|\delta\|_2 \},$$

where $\lambda' \in \mathbb{R}_{\geq 0}$ is a hyperparameter, using gradient descent on $\delta$. The term $\|\delta\|_2$ is designed to encourage the recourse $\delta$ to be small, which is often desirable in practice (we have excluded it from our formulation for simplicity). While this approach is generally effective, it can be very slow since we need to solve an optimization problem for each individual.

**Adversarial training.** We can also use adversarial training to compute recourse—i.e.,

$$\delta_x = \arg\min_{\delta \in \Delta} \nabla_x \ell(g_\theta(x), 1)^\top \delta.$$

This approach approximates the gradient descent approach, but can be computed much more efficiently. Furthermore, since our objective in Eq. 1 is designed to encourage this specific perturbation to provide recourse, it performs nearly as well as gradient descent when $f_\theta$ is trained with $\lambda > 0$.

**Linear approximation.** Finally, Ustun et al. [2019] propose an approach to compute recourse when $f_\theta(x) = \mathbb{1}(\beta^\top x \geq \beta_0)$ is a linear model. In this case, they compute $\delta_x$ using an integer linear program (ILP). For nonlinear models, we can instead use the linear approximation of $g_\theta$ near $x$. Letting $\delta_x = \mathcal{A}(x; \beta_0, \beta)$ be the recourse generated by their algorithm, and using the Taylor expansion $g_\theta(x') \approx g_\theta(x) + \nabla_x g_\theta(x)^\top (x' - x)$, we can use their approach to compute the recourse

$$\delta_x = \mathcal{A}(x; \theta_0 - g_\theta(x), \nabla_x g_\theta(x)).$$

However, this approach only works well when $g_\theta$ is approximately linear as a function of $x$; otherwise, the Taylor approximation may be poor and $\delta_x$ may not satisfy the desired condition $f_\theta(x + \delta_x) = 1$.

### 3.3   Step 2: Guaranteeing Recourse

Finally, we describe how we choose $\theta_0$ to ensure $f_\theta$ provides recourse with high probability. Note that $\theta_0$ also controls the fraction of individuals given a positive outcome *without* the need for recourse; thus, we choose $\theta_0$ to optimize the performance of $f_\theta$ subject to a constraint that $f_\theta$ is PARE.

**Background: PAC confidence sets.** We build on work constructing PAC confidence sets [Park et al., 2020]. Given $x \in \mathcal{X}$, they construct a model $\tilde{h}_{\theta,\tau} : \mathcal{X} \to \mathcal{P}(\mathcal{Y})$ (where $\mathcal{P}$ is the power set) that returns the set of all labels $y$ with score above threshold $\tau \in \mathbb{R}$—i.e.,

$$\tilde{h}_{\theta,\tau}(x) = \{y \in \mathcal{Y} \mid h_\theta(x, y) \geq \tau\} \quad \text{where} \quad h_\theta(x, y) = \begin{cases} g_\theta(x) & \text{if } y = 1 \\ 1 - g_\theta(x) & \text{if } y = 0. \end{cases}$$

Given $\epsilon \in \mathbb{R}_{>0}$, we say $\tau$ is *approximately correct* if

$$\mathbb{P}_{p(x,y)}[y \in \tilde{h}_{\theta,\tau}(x)] \geq 1 - \epsilon,$$

i.e., $\tilde{h}_{\theta,\tau}(x)$ contains the true label for $x$ with high probability over $p(x, y)$. Note that $\tau = 0$ satisfies this condition, since $\tilde{h}_{\theta,0}(x) = \{0, 1\}$ for all $x$. The goal is to choose $\tau$ as large as possible while ensuring approximate correctness.

Park et al. [2020] proposes an estimator $\hat{\tau}$ that takes as input (i) the pretrained model $g_\theta : \mathcal{X} \to [0, 1]$, (ii) a calibration dataset $Z \subseteq \mathcal{X} \times \mathcal{Y}$ of i.i.d. samples $(x, y) \sim p$, and (iii) confidence levels $\epsilon, \alpha \in \mathbb{R}_{>0}$, and constructs a threshold $\hat{\tau}(Z) \in [0, 1]$ that is *probably approximately correct (PAC)*:

$$\mathbb{P}_{p(Z)}[\hat{\tau}(Z) \text{ is approximately correct}] \geq 1 - \alpha. \tag{3}$$

In other words, $\hat{\tau}(Z)$ is approximately correct with probability at least $1 - \alpha$ according to $p(Z)$. Their approach leverages the fact that $\hat{\tau}(Z)$ is an estimator for a single parameter; thus, they can use learning theory to obtain PAC guarantees [Kearns et al., 1994].

**PARE models via PAC confidence sets.** Given a model $g_\theta : \mathcal{X} \to \mathbb{R}$, an algorithm $\mathcal{A} : \mathcal{X} \to \Delta$ for computing recourse, and a calibration set $Z \subseteq \mathcal{X} \times \mathcal{Y}$, our algorithm leverages PAC confidence sets to choose a threshold $\theta_0 = \hat{\theta}_0(Z)$ that ensures the resulting model $f_{\hat{\theta}}$ satisfies the PARE constraint.

First, our algorithm uses the PAC confidence set algorithm to construct the new calibration dataset

$$Z' = \{(x + \mathcal{A}(x), 1) \mid (x, y) \in Z\}.$$

Intuitively, $Z'$ says that the "correct" label for every input $x + \mathcal{A}$ should be 1—i.e., the recourse constructed by $\mathcal{A}$ should satisfy $f_\theta(x + \mathcal{A}(x)) = 1$.

| Metrics | Adult | | Compas | | Bail | | German | |
|---|---|---|---|---|---|---|---|---|
| | Baseline | Ours | Baseline | Ours | Baseline | Ours | Baseline | Ours |
| **Performance** | | | | | | | | |
| F1 score | **0.697** | 0.636 | **0.739** | 0.717 | **0.775** | 0.760 | **0.447** | 0.419 |
| Accuracy | **0.830** | 0.787 | **0.667** | 0.565 | **0.643** | 0.629 | **0.600** | 0.527 |
| Precision | **0.621** | 0.555 | **0.655** | 0.561 | **0.646** | 0.644 | **0.364** | 0.317 |
| Recall | **0.799** | 0.752 | 0.850 | **0.991** | **0.968** | 0.930 | 0.583 | **0.638** |
| **Recourse neg** | | | | | | | | |
| Linear approx. | **0.220** | 0.053 | **0.156** | 0.068 | **0.102** | 0.029 | **0.204** | 0.086 |
| Gradient desc. | 0.210 | **0.496** | 0.579 | **1.000** | 0.317 | **1.000** | 0.804 | **0.864** |
| Adversarial train. | 0.007 | **0.498** | 0.000 | **0.967** | 0.000 | **0.993** | 0.127 | **0.968** |
| **Recourse all** | | | | | | | | |
| Linear approx. | **0.453** | 0.328 | 0.773 | **0.982** | 0.957 | **0.981** | **0.607** | 0.593 |
| Gradient desc. | 0.461 | **0.661** | 0.883 | **1.000** | 0.970 | **1.000** | 0.890 | **0.920** |
| Adversarial train. | 0.321 | **0.659** | 0.722 | **0.999** | 0.952 | **0.999** | 0.517 | **0.980** |

Table 1: Performance and recourse for the baseline model ($\lambda = 0$) and the model trained with our algorithm ($\lambda = 0.8$), and for each of the three algorithms for computing recourse in Section 3.2. We show mean results across 3 random data splits and bold the higher value between the baseline and our algorithm.

Then, our algorithm constructs $\tilde{h}_{\theta,\hat{\tau}(Z)}$ using $g_\theta$, $Z'$, and the given $\epsilon, \alpha$. The PAC guarantee in Eq. 3 says that with probability at least $1 - \alpha$ over $p(Z)$, we have

$$\mathbb{P}_{p(Z)}\left[\mathbb{P}_{p(x,y)}[y' \in \tilde{h}_{\theta,\hat{\tau}(Z')}(x')] \geq 1 - \epsilon\right] \geq 1 - \alpha, \tag{4}$$

where $(x', y') \in Z'$ is the example constructed from $(x, y) \in Z$ as described above. Note that the outer probability is over $p(Z)$ since $Z'$ is a deterministic function of the random variable $Z$. Plugging in the definitions $x' = x + \mathcal{A}(x)$ and $y' = 1$, Eq. 4 becomes

$$\mathbb{P}_{p(Z)}\left[\mathbb{P}_{p(x,y)}[1 \in \tilde{h}_{\theta,\hat{\tau}(Z')}(x + \mathcal{A}(x))] \geq 1 - \epsilon\right] \geq 1 - \alpha,$$

and plugging in the definition of $\tilde{h}_{\theta,\tau}$, it becomes

$$\mathbb{P}_{p(Z)}\left[\mathbb{P}_{p(x,y)}[g_\theta(x + \mathcal{A}(x)) \geq \hat{\tau}(Z')] \geq 1 - \epsilon\right] \geq 1 - \alpha. \tag{5}$$

Then, our algorithm returns $\hat{\theta}_0(Z) = \hat{\tau}(Z')$ (with given parameters $\theta$ as the remaining parameters). Since Eq. 5 is equivalent to the PARE condition, we have:

**Theorem 3.1.** $\hat{\theta}$ *satisfies the PARE condition.*

## 4 Experiments

We evaluate our approach and show how it can effectively improve recourse rates while preserving accuracy. We also demonstrate how it can improve the correctness and robustness of recourses. In Appendix B, we provide additional results on how our approach affects fairness of models, as well as results on an NLP task with discrete covariates to demonstrate the flexibility of our framework.

### 4.1 Experimental Setup

**Datasets.** We use four real-world datasets. The first contains **adult income** information from the 1994 United States Census Bureau [Dua and Graff, 2017]. It includes information about adults' demographics, education, and occupations. Each adult is labeled as making below or $> \$50K$ a year, which can be thought of as a proxy for whether an individual will be able to repay a loan or not. The second contains information collected by Propublica about criminal defendants' **compas recidivism scores** [Angwin et al., 2016]. This dataset includes information about defendants' demographics and crimes, and each defendant is labeled as having either a high or low likelihood of reoffending,

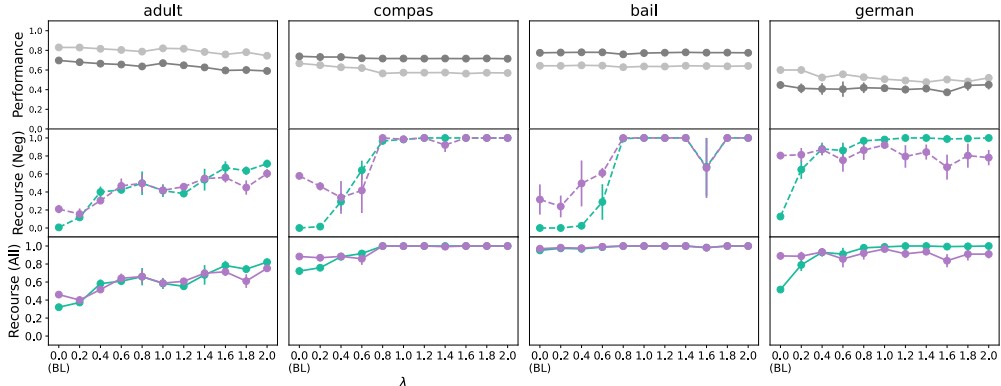

Figure 1: How performance and recourse vary with $\lambda$; $\lambda = 0$ is the baseline and $\lambda > 0$ is our approach. We plot means and standard errors across 3 random data splits. The first row shows performance metrics; the second and third show recourse metrics. *Performance:* ■ $F_1$ ■ Accuracy. *Recourse Algorithm:* ■ "gradient descent" ■ "adversarial training".

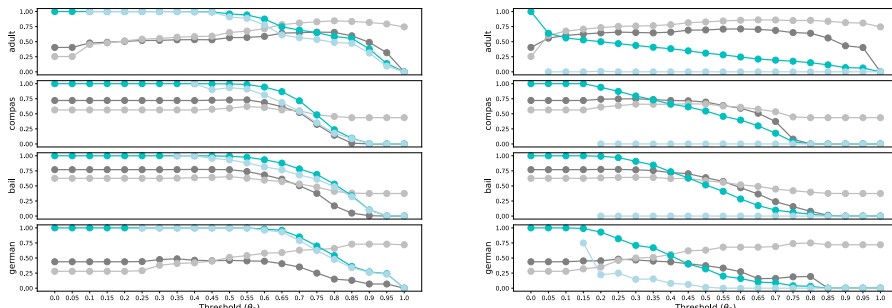

Figure 2: How performance and recourse vary with $\theta_0$, for our approach ($\lambda = 0.8$, left) and the baseline ($\lambda = 0$, right). *Performance:* ■ $F_1$ ■ Accuracy. *Recourse:* ■ Recourse neg ■ Recourse all.

as measured by the compas assessment tool. The third dataset represents **bail outcomes** from two different U.S. state courts from 1990-2009 [Schmidt and Witte, 1988]. It includes information about individuals' criminal histories and demographics. Each defendant is labeled as having a high or low risk of recidivism. The fourth dataset is the **german credit** dataset [Dua and Graff, 2017], which contains individuals' demographic, personal, and financial information. Each applicant is labeled as either having high or low credit risk.

We standardize all continuous features to have mean 0 and variance 1. We randomly split each dataset into 80% train and 20% validation sets. We use 3 random data splits and report the mean across splits for the rest of this section, unless otherwise specified. For compas and adult, we hold out 500 examples from the validation set to form a test set; for german, we hold out 100 examples. For bail, we evaluate on a 500-instance subset of the test set. All results are reported on the test set.[4]

**Models.** All models are neural networks with 3 100-node hidden layers, dropout probability 0.3, and tanh activations. For evaluation, we choose the epoch achieving the highest validation $F_1$ score.

**Parameters.** We experimented with $\lambda$ values between 0.0 to 2.0 in increments of 0.2 and found that $\lambda = 0.8$ provided the best tradeoff between $F_1$ score and recourse rate across all datasets; we evaluate how our results vary with $\lambda$ below. For $\Delta$, we choose the set of actionable features (i.e., features that can be changed as part of the recourse) based on the dataset (two features for adult, bail, german; one for compas); we set $\delta_{\max} = 0.75$ after standardizing features. (See Appendix B.2 for experiments investigating the effect of varying values of $\delta_{max}$). We also include domain-specific linear constraints in $\Delta$—e.g., for the adult dataset, recourse can only require that hours worked increases. See Appendix A.3 for more details about our choices of $\Delta$.

---

[4]Our processed datasets have sizes: adult $\approx 32.5K$, compas $\approx 6K$, bail $\approx 8K$, german $\approx 1K$.

We choose the decision threshold $\theta_0$ to maximize F1 score rather than to obtain PAC guarantees, since our goal is to understand the tradeoffs between model performance and recourse rates with a fixed underlying predictive model $f_\theta$. We evaluate the effects of rigorously choosing $\theta_0$ in Section 4.2.

**Baselines.** To the best of our knowledge, our approach is the first to train models with the objective of providing recourse at a higher rate. Thus, we compare to a baseline that omits our recourse loss—i.e., $\lambda = 0.0$. For our approach and this baseline, we evaluate the performance of each of the three different algorithms for computing recourse described in Section 3.2. We use the alibi implementation [Klaise et al., 2019] of the gradient descent algorithm for computing recourse [Wachter et al., 2017] and set the initial value of the hyperparameter $\lambda' = 0.001$. We use LIME [Ribeiro et al., 2016a] as our linear approximation method with the approach proposed by Ustun et al. [2019].

**Metrics.** We evaluate our approach and the baselines with the following metrics: (i) standard performance metrics of accuracy and $F_1$ score, (ii) "recourse neg," the proportion of instances $x$ with negative original outcomes that receive recourse such that $f(x) = 0$ but $f(x + \delta^*) = 1$, and (iii) "recourse all," the proportion of all instances $x$ such that *either* $f(x) = 1$ *or* $f(x) = 0$ but $f(x + \delta^*) = 1$. Metric (iii) is most useful for measuring rate of positive outcomes, since we want to include individuals who are originally assigned a positive outcome.

### 4.2 Efficacy of Our Framework

In Table 1, we show the performance and recourse rates of models trained with our approach and baseline models. Overall, our approach significantly improves recourse without sacrificing performance. Across datasets, models trained using our approach offer recourse at significantly higher rates than the baseline model, for both the "adversarial training" and "gradient descent" approaches to computing recourse. We do not observe this trend when using "linear approximation" to compute recourse; in this case, both the baseline and our approach perform poorly. We believe this effect can be explained by poor LIME approximations of $f_\theta$, which are exacerbated by adversarial training since it increases the nonlinearity of $f_\theta$. Figure 1 shows how these results vary with $\lambda$: $F_1$ scores and accuracies are relatively stable as a function of $\lambda$.

In Figure 2, we plot how these results vary with $\theta_0$ (for classifiers trained with $\lambda = 0$ and $\lambda = 0.8$ on a single data split), using the "adversarial training" method of computing recourse.[5] High values of $\theta_0$ lead to lower recourse rates in all cases. The curves for performance are similar for the baseline and for our approach, but the decline in recourse values begins at lower thresholds in the case of the baseline. Thus, our approach improves recourse for most choices of $\theta_0$ without sacrificing performance. The trade-off between recourse and performance depends on the dataset. For adult, performance increases while recourse decreases since the majority label in this dataset is negative, whereas for bail and compas, performance and recourse both decrease as $\theta_0$ increases since the majority label is positive.

**Performance under PARE guarantees.** Next, we show that we can often obtain PARE guarantees without significantly reducing performance. We compare the performance of choosing the decision threshold $\theta_0$ to maximize $F_1$ score to that of $\theta_0$ chosen to obtain PARE classifiers. Specifically, for the latter, we compute $\theta_0$ using the approach described in Section 3.3 with parameters $\epsilon = \alpha = 0.05$. Then, we evaluate models at thresholds in 10 equally spaced increments from 0 to the upper bound and fix $\theta_0$ to maximize F1 score. For these experiments, we use the "adversarial training" algorithm to compute recourse; we observed similar trends for the "gradient descent" algorithm.

Results are shown in Table 2. In all cases, choosing $\theta_0$ to satisfy the PARE condition yields a classifier that returns recourse at a rate $\geq 1 - \epsilon = 0.95$, which validates our theoretical guarantees. Furthermore, on all three datasets, the $F_1$ score does not significantly decrease when imposing the PARE condition. We do see a decrease in $F_1$ score for the baseline model on the adult dataset, but the decrease for our model is smaller, suggesting that our end-to-end framework of training models and fixing $\theta_0$ is successful at guaranteeing recourse without a big drop in accuracy.[6]

---

[5]Note that the "recourse neg" values are low for $\lambda = 0$ because we use the "adversarial training" method to compute recourse, which builds on a fast linear approximation and thus does not exhaustively find recourses. We use this method instead of the more effective "gradient descent" method for efficiency, since the latter is less efficient and would require recomputing recourses for each threshold.

[6]We can obtain a weaker theoretical guarantee at a smaller cost in performance. For instance, applying PARE to our model in Table 2 with $\epsilon = \alpha = 0.25$ results in an F1 score of 0.613 on the adult dataset.

|           | **Adult** | | **Compas** | | **Bail** | | **German** | |
|-----------|-------|----------|-------|----------|-------|----------|-------|----------|
|           | $F_1$ | Recourse | $F_1$ | Recourse | $F_1$ | Recourse | $F_1$ | Recourse |
| BL + $F_1$ max | 0.697 | 0.321 | 0.739 | 0.722 | 0.775 | 0.952 | 0.447 | 0.517 |
| **BL + PARE** | 0.400 | 0.981 | 0.722 | 0.972 | 0.777 | 0.996 | 0.442 | 0.997 |
| Ours + $F_1$ max | 0.636 | 0.659 | 0.717 | 0.999 | 0.760 | 0.999 | 0.419 | 0.980 |
| **Ours + PARE** | 0.526 | 0.974 | 0.721 | 0.999 | 0.776 | 0.999 | 0.457 | 1.000 |

Table 2: Impact of choosing decision threshold $\theta_0$ to satisfy the PARE constraint. We show $F_1$ scores for the baseline model and the model trained with our algorithm using two methods of determining $\theta_0$: (i) maximize the $F_1$ score, and (ii) guarantee that the model satisfies the PARE constraint (+PARE, bolded). For the baseline model (BL), $\lambda = 0$; for our model (Ours), $\lambda = 0.8$.

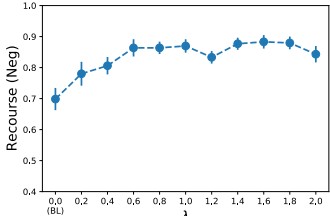

Figure 3: "Recourse neg" for the german dataset using the causal recourse computing algorithm proposed by Karimi et al. [2020b]. We show means and standard errors across 20 random data splits for varying values of $\lambda$.

**Groundedness of recourses.** One key question is whether the recourses of models trained with our approach are *grounded* in reality—i.e., whether they are plausible modifications of the ground truth label. For instance, if an individual is denied a loan, they should be given a recourse such that if they take the specified action, they actually increase their likelihood of paying back the loan; an individual may increase their income and be given a loan, but still fail to repay it. We want to ensure that our training approach increases the rate of recourses that obey causal relationships in the world, rather than recourses that exploit spurious correlations between features. Directly evaluating groundedness of recourses is challenging, since we do not know the ground truth labels for suggested recourses. Thus, we evaluate whether they are *causally grounded* and *in-distribution*.

First, we measure the rate at which causally grounded recourses are offered by models trained with our approach. We apply the algorithm for computing causal recourses (CR) introduced by Karimi et al. [2020b] to evaluate whether our proposed training algorithm improves the rate at which causally grounded recourses are offered. Because CR requires access to an underlying structural causal model (SCM) of the world, we only experiment with the german dataset, for which Karimi et al. [2020b] provide an associated SCM. We measure the proportion of test instances $x$ for which the CR algorithm computes a valid recourse that satisfies the constraint that perturbations be bounded by $\delta_{max}$—i.e. "recourse neg".[7] As shown in Figure 3, with increasing $\lambda$, the rate at which recourses are by the CR algorithm increases. This finding suggests that our training algorithm encourages the existence of causally grounded recourses.

Second, we evaluate whether the recourses offered by models trained with our approach are in-distribution with respect to the original training data. In line with prior work [Slack et al., 2020], we train classifiers to distinguish between original data instances and recourses computed by the "gradient descent" algorithm for models trained with our approach ($\lambda = 0.8$). In Table 3, we report the accuracies of these classifiers on a held out test set. The low classifier accuracies across all datasets indicate that these recourses are indistinguishable from original data instances. Thus, our framework encourages the existence of recourses that are in-distribution to the original data.

**Robustness.** Another key question is whether the recourses generated using our approach are *robust*—i.e., whether small changes to the recourse result in valid recourses. For instance, if an individual

---
[7]We select hyperparameters for the CR algorithm that maximize "recourse neg" on the train set.

|                      | Adult | Compas | Bail | German |
|----------------------|-------|--------|------|--------|
| Neural Network       | 0.54  | 0.56   | 0.52 | 0.51   |
| Random Forest        | 0.53  | 0.55   | 0.51 | 0.51   |
| Logistic Regression  | 0.51  | 0.48   | 0.48 | 0.47   |

Table 3: Accuracies of classifiers trained to distinguish between original data instances and recourses computed using the "gradient descent" algorithm for computing recourse for models trained on a single random data split with our approach ($\lambda = 0.8$).

| Robustness Exp. | Recourse Alg. | Adult | | Compas | | Bail | | German | |
|-----------------|---------------|-------|------|--------|-------|------|-------|--------|-------|
|                 |               | BL    | Ours | BL     | Ours  | BL   | Ours  | BL     | Ours  |
| **Recourse**    | Grad. desc.   | 1.000 | 0.865| 1.000  | 1.000 | 1.000| 1.000 | 0.949  | 0.898 |
|                 | Advers. train.| 1.000 | 0.841| 1.000  | 1.000 | 1.000| 1.000 | 0.857  | 1.000 |
| **Model**       | –             | 0.976 | 0.926| 0.980  | 1.000 | 0.994| 0.996 | 0.970  | 0.920 |

Table 4: The first row shows the percentage of recourses found that are robust to noise. The second row shows the percentage of test inputs that are robust to noise in the recourse dimensions. We show results for models trained with varying $\lambda$ ($\lambda = 0$ indicates the baseline, and $\lambda = 0.8$ indicates our approach) on a single data split. For each model, we show results for the "gradient descent" and "adversarial training" algorithms for computing recourse in Section 3.2.

increases their income by more (or even slightly less) than the amount suggested in the recourse, they would expect to still be provided with a positive decision.

For each computed recourse $\delta$, we compute a noisy recourse $\delta'$ by adding i.i.d. Gaussian noise to each actionable feature of $\delta$—i.e., $\delta_i' = \delta_i + \mathcal{N}(0, 0.1)$. We consider a recourse $\delta$ *robust* if its noisy recourse $\delta'$ is valid—i.e. if $f(x + \delta') = 1$. In the top row of Table 4, we report the percentage of recourses found that are robust for a baseline model and model trained with our adversarial approach. As shown, our training approach does not significantly reduce the robustness of recourses: There is a slight drop in robustness for the adult dataset and for the "gradient descent" recourse algorithm for the german dataset; however, for compas and bail, recourses remain robust.

**Effect on classifier brittleness.** We also investigate the effect of our adversarial training approach on model brittleness. In particular, we measure how sensitive models trained with our adversarial algorithm are to noises in the recourse dimensions, as compared to baseline models. We add i.i.d. Gaussian noise, as described above, to the actionable dimensions of *original inputs*, and compute the proportion of test instances for which the trained model is robust to this noise. As shown in the bottom row of Table 4, our training approach does not significantly increase model brittleness—we observe a small drop in model robustness in the recourse dimensions for the adult and german datasets and a small increase in robustness for the compas and bail datasets. These results suggest that our training approach effectively ensures recourse without rendering classifiers brittle.

## 5 Conclusion

We have proposed a novel algorithm for training models that are guaranteed to provide recourse for reversing adverse outcomes with high probability. Our experiments show that our algorithm trains models that provide recourse at high rates without sacrificing accuracy compared to traditional learning algorithms. Future work includes extending our techniques beyond binary classification.

## Acknowledgements

We would like to thank the anonymous reviewers for their insightful feedback. This work is supported in part by the NSF awards #IIS-2008461, #IIS-2040989, and #CCF-1910769, and research awards from the Harvard Data Science Institute, Amazon, Bayer, and Google. The views expressed are those of the authors and do not reflect the official policy or position of the funding agencies.

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
