# A Experiment Details

## A.1 Training Details

We train our models with the ADAM optimizer using a learning rate of 0.002. For adult, compas, and bail, we train for 15 epochs using a batch size of 15. For german, train for 50 epochs and use a batch size of 30.

## A.2 Data Processing

We use the following features for training our models.

- **Adult:** "age," "education-num," "capital-gain," "capital-loss," "hours-per-week," "race," "native-country," "marital-status," "sex."
- **Bail:** "WHITE," "ALCHY," "JUNKY," "SUPER," "MARRIED," "FELON," "WORKREL," "PROPTY," "PERSON," "MALE," "PRIORS," "SCHOOL," "RULE," (i.e., the number of prison rule violations reported during the sample sentence) "AGE," "TSERVD," "FOLLOW" (i.e., length of the followup period)
- **Compas:** "age," "priors_count," "length_of_stay," "days_b_screening_arrest," "sex," "race," "c_charge_degree."
- **German:** "gender," "age", "duration," (i.e., repayment duration of the credit), and "personal_status_sex" (i.e. credit given by the bank).

## A.3 Choices of $\Delta$

Our approach allows for customization of actionable features and constraints on their values. Here, we describe the actionable features and constraints we used in our experiments:

- **Adult:** The actionable features are (i) education level, and (ii) number of hours worked per week. We require that education level can only increase.
- **Bail:** The actionable features are: (i) education level and (ii) the number of prison rule violations reported during the sample sentence. We require that education level can only increase.
- **Compas:** The actionable feature is the number of prior crimes.
- **German:** Age and credit given by the bank. We require that age can only increase.[8]

Note that "past" features like the number of prison rule violations and number of prior crimes can be treated as actionable if the individual can wait and re-apply for an outcome.

# B Additional Experiments

## B.1 Classifier Robustness to Realistic Noise

In addition to leveraging Gaussian noise to assess the brittleness of classifiers trained with our approach (see Section 4.2), we also experimented with other kinds of perturbations. Specifically, we experimented with: (A) the Natural Adversarial Examples (NAE) approach [Zhao et al., 2018], which employs GANs to generate adversarial examples that lie on the data manifold, (B) a variant of (A) that leverages GANs to generate small random perturbations that lie on the data manifold but does not explicitly optimize for the perturbations to have different labels than the original instances, and (C) the Deepfool approach [Moosavi-Dezfooli et al., 2016], which is an iterative gradient-based approach to generate adversarial examples for a given input sample.

We add these perturbations to the actionable dimensions of original inputs and compute the proportion of test instances for which the classifiers are robust to the perturbations. We find that the difference in robustness (measured the same way as described in Section 4.2) between classifiers produced by our framework ($\lambda = 0.8$) and baseline classifiers produced without our recourse loss ($\lambda = 0$) is $\leq 0.03$

---

[8]We choose these features based on the set-up of Karimi et al. [2020b]

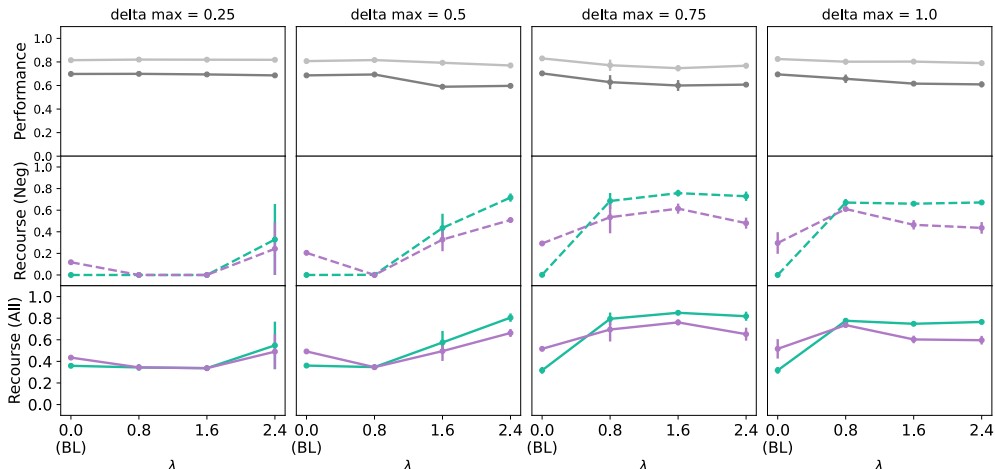

Figure 4: How performance and recourse vary with $\delta_{\max}$ for the adult dataset. *Performance:* ■ $F_1$ ■ Accuracy. *Recourse Algorithm:* ■ "gradient descent" ■ "adversarial training".

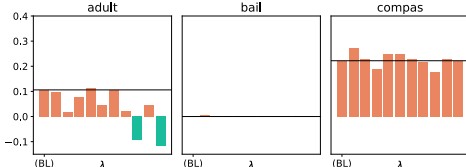

Figure 5: Recourse disparity vs. $\lambda$, using the "recourse all" metric. A positive value (■) indicates that whites receive recourse at a higher rate than non-whites—i.e. a racial disparity exists; a negative value (■) indicates the reverse. We show the baseline (BL) model (i.e., $\lambda = 0$; also the black horizontal line) and for models trained using our approach with $\lambda \in [0.2, 0.4, ..., 2.0]$.

across all datasets. Our results indicate that the classifiers produced by our framework are comparable in terms of their robustness to baseline classifiers even with these perturbation techniques.

## B.2 Effect of $\delta_{\max}$

We also evaluate the sensitivity of our results to different choices of $\delta_{\max}$ on the adult dataset. As shown in Figure 4, our training approach improves recourse rates without noticeably reducing performance for different values of $\delta_{\max}$.

## B.3 Recourse Disparities for Minorities

We assess the effect of our adversarial training objective on recourse disparities for whites (which we consider to be the majority subpopulation) vs. non-whites (which we consider to be the minority subpopulation). In particular, we investigate whether our training objective worsens recourse disparities. For each of our three datasets, we compare the disparities in "recourse all" values of whites and non-whites between a baseline model (i.e., $\lambda = 0$) and models trained with our approach (i.e., $\lambda > 0$). We choose a threshold $\theta$ to fix precision at a value of $0.65$, and compute *recourse all* separately for the majority and minority subpopulations. In this experiment, we use the "gradient descent" algorithm to compute recourse since it finds recourse at the highest rate.

Results are in Figure 5. Overall, we find that our training approach does not worsen recourse disparities. For the compas dataset, we find that our training approach does not worsen the existing disparity in recourse rates offered by the baseline model to whites and non-whites. For the bail dataset, there is no disparity in recourse rates for whites and non-whites offered by the baseline model, and our training approach does not introduce one. For the adult dataset, our training approach in fact *reduces* disparity: The baseline models provide recourse to whites at a higher rate than to non-whites,

| Metrics | Baseline $(\lambda = 0)$ | Adversarial $(\lambda = 0.25)$ |
|---|---|---|
| **Performance** | | |
| F1 score | 0.881 | 0.864 |
| Accuracy | 0.875 | 0.862 |
| Precision | 0.926 | 0.878 |
| Recall | 0.839 | 0.850 |
| **Recourse** | | |
| Recourse: neg | 0.239 | **0.950** |
| Recourse: all | 0.656 | **0.976** |

Table 5: Impact of our approach on a sentiment classifier; $\theta_0$ is chosen to maximize the $F_1$ score.

| Feature | $x$ | $x + \delta$ | $\delta$ |
|---|---|---|---|
| education level | 13.0 | 14.397 | **+ 1.397** |
| hours per week worked | 40.0 | 42.921 | **+ 2.921** |

| Feature | $x$ | $x + \delta$ | $\delta$ |
|---|---|---|---|
| number of prior crimes | 2.0 | 0.667 | **–1.333** |

Table 6: Example recourse computed using the "gradient descent" algorithm [Wachter et al., 2017] for a model trained with our approach ($\lambda = 0.8$) for a test instance from the adult dataset (top) and from the compas dataset (bottom).

while models trained with our approach reduce the magnitude of the disparity or even reverse the disparity (green bars) for varying values of $\lambda$.

## B.4 NLP Case Study

We investigate whether our approach can be applied to another domain; in particular, we apply our approach to sentiment classification. We use the Stanford Sentiment Treebank dataset, which contains movie reviews labeled by sentiment [Socher et al., 2013]. We use the train/dev/tests in the dataset. We treat positive sentiment as the positive outcome. We train a model consisting of a linear layer on top of a pretrained BERT model [Devlin et al., 2019, Wolf et al., 2019], which we finetune for two epochs with a learning rate of $2e - 5$ and a linear learning rate scheduler.

Since the covariates are discrete, we cannot use our choice of $\Delta$ or our approach to solving for $\delta^*$ in Eq. 2. Instead, we choose $\Delta$ to be the set of perturbations obtained by replacing a single noun or adjective in the original sentence with one of its antonyms (obtained from a thesaurus). For each $x$, we approximate Eq. 2 as $\delta^* = \arg\min_{\delta \in \hat{\Delta}} \ell(g_\theta(x \oplus \delta), 1)$, where $\hat{\Delta} = \{\delta_1, ..., \delta_n\}$ is a set of candidates from $\Delta$ (we choose $n = 10$), and where $x \oplus \delta$ is the result of replacing a word in $x$ with its antonym encoded by $\delta$. For instance, for [*There's no emotional pulse to Solaris*], $\delta^*$ is [*There's no **unexcited** pulse to Solaris*].

Results on the test set are shown in Table 5. Our approach increases the rate at which recourses are given without noticeably decreasing performance, offering preliminary evidence that our approach can be applied to non-tabular data using different techniques for computing $\delta^*$.

## B.5 Examples of Recourses

We show examples of recourses generated using our approach in Table 6. The top recourse says the individual should increase their education level by 1.397 years and increase the number of hours worked per week by 2.921 to receive a positive outcome of approval for a loan. The bottom recourse says the individual should reduce their number of prior crimes by 1.333 to receive the positive outcome of low recidivism likelihood prediction.