# OpenReview forum: "Learning Models for Actionable Recourse"
_NeurIPS.cc/2021/Conference — NeurIPS 2021 Poster_

### Official Review · Reviewer_jpk7 · 2021-07-08

**Rating:** 6
**Confidence:** 1

**Summary:**

The author(s) proposed an adversarial training-based classification algorithm that guarantees affected individuals receive recourse with a high probability. The paper considers an interesting topic. The numerical experiments demonstrate the effectiveness of the proposed algorithm.

**Limitations And Societal Impact:**

I suggest the author(s) to discuss more about the motivations of learning classifiers with actionable recourses. In addition, the author(s) might want to adopt a counterfactual outcome framework to formulate their problems.

**Main Review:**

**Pros**

* The paper is written clearly and well-organised.

* The topic is interesting. The author(s) considered classification algorithms that ensure adversarial examples do exist.

**Cons**

* It remains unclear to me why it is useful to consider algorithms with actionable recourses in practice. It would be helpful if the author(s) could discuss this in more detail.

* The author(s) considered counterfactual explanations but did not adopt a causal framework for problem formulation. It would be better if a counterfactual outcome framework is used to present the problem, the methods and the results.

**Time Spent Reviewing:**

2h

---

> ### Author Response · Authors · 2021-08-11
> **Response to Reviewer jpk7**
>
> Thank you for the review! Below are responses to specific comments from the reviewer:
>
> **Re: “It remains unclear to me why it is useful to consider algorithms with actionable recourses in practice. It would be helpful if the author(s) could discuss this in more detail.”**
>
> As machine learning (ML) models are increasingly being deployed to make a variety of consequential decisions ranging from hiring decisions to loan approvals, it becomes critical to provide individuals who have been adversely impacted by algorithmic decisions with a means for recourse (i.e., a way to achieve a positive outcome). For example, suppose a person is denied a loan by a machine learning model; it would be critical to provide them with a way to modify features under their control so that they can qualify for a loan when they reapply (an example recourse could take the following form: “increase your income by X dollars and savings by Y dollars”). Otherwise, this individual has no agency over a crucial decision that greatly affects their life. Ustun et al., (2019) and Wachter et al. (2017) provide thorough arguments to highlight the need for recourses. We will clarify this motivation in our introduction.
>
> **Re: “The author(s) considered counterfactual explanations but did not adopt a causal framework for problem formulation. It would be better if a counterfactual outcome framework is used to present the problem, the methods and the results.”**
>
> We would like to clarify some terminology here. In this work, we adopted the terminology and conventions introduced by Wachter et. al. (2017). According to these conventions, despite what the name suggests, “counterfactual explanations” do not relate to causal frameworks. In particular, the notion of counterfactual explanations defined by Wachter et. al. ignores the casual structure of the underlying data distribution.
>
> On the other hand, the notion of counterfactual explanations put forth by Karimi et. al. (2020b) leverages causal graphs to obtain recourses. In fact, Karimi et al. (2020b) suggested using the alternative term “contrastive explanation” to refer to explanations generated by Wachter et. al. (2017) which are not causally grounded; we are happy to adopt this terminology to avoid confusion.
>
> Note that our results which highlight the effect of our training approach on the rates of causally grounded recourses (Figure 3, Section 4.1) build on this framework by Karimi et. al. (2020b). However, our framework itself is generic enough to be applicable to both causally grounded and non-causally-grounded recourses.
>
> We will clarify this in the final version.
>
> We hope we addressed all your questions/concerns/comments adequately. In light of our clarifications, please consider increasing your score to accept. Please let us know if we can provide any further details and/or clarifications.

---

### Official Review · Reviewer_tuPn · 2021-07-15

**Rating:** 6
**Confidence:** 2

**Summary:**

The paper develops an adversarial training framework for training models such that they are likely to have recourse options (small changes to the input which lead to switching decisions from 0 to 1); and a method for providing a PAC-inspired guarantee that an algorithm has recourse options with high probability.

**Limitations And Societal Impact:**

See [setup concerns] above.


**Main Review:**

I'm going to separate my review into general concerns about the setup, and then comments on the technical merits of the paper that assume that the general concerns are mitigated. The technical details of this paper are strong but they require that the setup makes sense (and that's where most of my concerns are) - so it's helpful to separate the two.

**[Setup  concerns]** The key premise of this paper is that negative automated decisions require "small" actionable recourse with high probability. They give the example of an "applicant denied a loan, a counterfactual explanation might be: “to get a loan, increase your income by $1,000”", and want such recourse to occur with high probability. Given the massive impact that some of these systems have on applicants (bail decisions, recidivism, etc.), this is a desirable property (although it is not obvious that most recourses should be "small"): if we're going to use machine learning to assist in such consequential decisions, at the very least we should provide a recourse avenue to those who get an undesirable prediction; and recourse based on counterfactual inputs has become a popular approach in the explainable AI community.

That said, I find the idea of using adversarial-style training (proposed in this paper) to ensure a model has a way of providing recourse worrying, because it asks to ensure that counterfactual inputs for recourse exist under the model without addressing whether or not such inputs *should* lead to a positive decision in reality? Overall very little attention is paid to whether, $f_\theta(x+\delta) = f(x+\delta)$  - i.e. whether the proposed recourse would be labeled as true under some true labelling process, $f$? This seems very problematic given that adversarial examples are usually used to highlight the arbitrariness of machine learning decisions when input data differs by small amounts from the training distribution. If the recourse suggestions have no grounding in reality---and the proposed adversarial training procedure seems likely to make this problem worse---then I'd worry that you're asking people to make arbitrary (and likely costly) changes to their lives in order to satisfy a model (e.g. it's good that it's possible to enforce constraints to ensure the procedure doesn't suggest decreases to income to qualify for a loan, but such constraints shouldn't be necessary if the proposed recourse set had any grounding in reality).

The results shown in the recourse correctness are give some defence against this critique, but I don't know that I share their conclusions - in Figure 3, $\lambda = 0$ and  $\lambda = 2.$ are not significantly different from each other, and generally there is a lot of overlap in the confidence intervals, so I'm not sure you can conclude that the training algorithm "training algorithm encourages the existence of causally grounded recourses," especially from only a single dataset.

This concern is the primary reason for my vote to reject. That said, this isn't my area of research, so I'll happily increase my score if the rebuttal and / or other reviewers convince me this isn't such an issue, hereafter I assume that these concerns are not an issue when evaluating the technical contribution...

**[Technical merit]** There are two parts to the approach - the adversarial training, and an approach to choosing the decision threshold such that we ensure a model "Probably Approximately has REcourse" (PARE; analogous to PAC guarantees). Both leverage existing techniques but were very well executed, and I particularly enjoyed the way that the paper uses PAC confidence sets [Park et al 2020] to ensure recourse with high probability. I could imagine such guarantees being required by regulators.

The experiments show that the method is works as intended and provides recourse options that are relatively robust.

**Time Spent Reviewing:**

5

---

> ### Author Response · Authors · 2021-08-11
> **Response to Reviewer tuPn**
>
> We appreciate the thorough review and detailed suggestions for improving our paper! Below, we respond to specific questions raised by the reviewer.
>
> **Re: “The results shown in the recourse correctness are give some defence against this critique, but I don't know that I share their conclusions - in Figure 3, λ=0 and λ=2 are not significantly different from each other, and generally there is a lot of overlap in the confidence intervals, so I'm not sure you can conclude that the training algorithm "training algorithm encourages the existence of causally grounded recourses," especially from only a single dataset.”**
>
> In response to the reviewer’s comment about Figure 3, we ran the CR experiments with more random seeds (20 instead of the original 3 presented in the paper). Thanks to these larger number of trials, we now observe clearer trends which demonstrate that our adversarial training improves CR rates (See https://imgur.com/5QL4tQA for new results with 20 seeds and error bars representing standard errors instead of standard deviations). We will include these new results in our paper.
>
> **Re: “I find the idea of using adversarial-style training (proposed in this paper) to ensure a model has a way of providing recourse worrying, because it asks to ensure that counterfactual inputs for recourse exist under the model without addressing whether or not such inputs should lead to a positive decision in reality?”**
>
> By addressing the concerns about our results in Figure 3 (in our previous comment), we hope that we have also addressed some of the questions regarding our recourses being grounded in reality. Below, we make several additional points to further address this concern.
>
> **Our recourses are not OOD and are realistic.** First, the reason why adversarial training produces perturbations that are “out-of-distribution” (OOD) is that for high-dimensional data such as images, making small changes to a large number of features results in a large change overall. This is why adversarial training produces perturbations that are out-of-distribution in case of high dimensional data. However, since we only allow our algorithm to modify one or two features, we expect the resulting recourses to be close to the training distribution which would in turn imply that the resulting recourses are more likely to be valid (i.e., hold true) under the true labeling process $f$ as well.
>
> To further confirm our intuition that our recourses are close to the training distribution, we carried out an additional experiment where we assessed if the original data instances and the recourses output by our framework are indistinguishable. To this end, we employed a strategy adopted by prior work (Slack et. al. 2020) and trained a classifier to distinguish between original data instances and the recourses output by our framework (i.e., $x + \delta$). If such a classifier achieves high accuracy (measured on a held-out test set), then it implies that our recourses are indeed OOD; otherwise, our recourses are not OOD and are in fact close to the training distribution and are grounded in reality.
>
> We experimented with three different types of classifiers, namely, deep neural networks (10 layers and 5 nodes in each layer), random forests (200 trees), and logistic regression. The accuracies of each of these classifiers across all of our datasets are given below:
>
> - Adult dataset: 0.54 (deep neural nets), 0.53 (random forests), 0.51 (logistic regression);
> - COMPAS dataset: 0.56 (deep neural nets), 0.55 (random forests), 0.48 (logistic regression);
> - Bail dataset: 0.52 (deep neural nets), 0.51 (random forests), and 0.48 (logistic regression);
> - German dataset: 0.51 (deep neural nets), 0.51 (random forests), 0.47 (logistic regression)
>
> The low classifier accuracies across all the datasets indicate that the recourses generated by our framework are indistinguishable from original data instances, thus confirming that our recourses are not OOD and are in fact close to the training distribution.
>
> **Additional strategies for ensuring realistic recourses.** There are a number of additional approaches that can be used in practice to ensure that the resulting recourses are valid under the true labeling process $f$; for instance, we can leverage causal knowledge (as we do in the results in Figure 3), or include additional training data (e.g., include those individuals in the training data who have acted upon the proposed recourse and whose outcomes are now available) and retrain the model in order to more effectively capture the true labeling process. We will include this discussion in our final version.
>
> **Importance of ensuring that recourses exist.** Finally, at a more fundamental level, we believe that imposing the existence of recourse on a model is an important constraint regardless of whether the resulting recourse remains valid under the true labeling process. In particular, if no recourse exists for an individual, it means that no reasonable change to their features can produce a positive outcome. For instance, in our loan example, it means that there is no way an applicant can modify features under their control to obtain a loan. We believe that such a model is fundamentally unfair, since the applicant has no agency over a decision that greatly affects their life. Thus, we believe that there is intrinsic value in ensuring that recourses exist.
>
> We will discuss all the aforementioned points in detail in our final version.
>
> We hope we addressed all your questions/concerns/comments adequately. In light of our clarifications, please consider increasing your score to accept. Please let us know if we can provide any further details and/or clarifications.

---

> > ### Comment · Reviewer_tuPn · 2021-09-01
> > **Response**
> >
> > Thanks for these additional experiments. I really like the tests for OOD that you include and have increased my score accordingly. Regarding, "Importance of ensuring that recourses exist" - I think that's a discussion worth including. Personally, I still think you're underweighting the negative consequences of granting loans to people who really can't afford them (under a true labelling process)---risky lending was the key driver of the financial crisis of 2008, and pushing who don't have the capacity to pay further into debt does not necessarily make them better off---but this is a debate that goes beyond the scope of the paper. I'm fine with stating that PARE is a desirable property in many situations (it is!) and then just alerting the reader to some of these considerations.

---

> > > ### Author Response · Authors · 2021-09-07
> > > **Thank you and response to follow up comments**
> > >
> > > We are very glad to hear that the reviewer found our rebuttal helpful. Thank you so much for increasing your score.
> > >
> > > We will definitely include the new results with the OOD tests as well as a discussion on the "Importance of ensuring that recourses exist" in the final version. We will also discuss all the relevant considerations as per your suggestions.

---

### Official Review · Reviewer_7Srb · 2021-07-16

**Rating:** 7
**Confidence:** 3

**Summary:**

The paper proposes the idea of training machine models that are effectively recourse ready. The authors propose leveraging adversarial training and PAC confidence sets to train models that "theoretically guarantee recourse". The experiments with four datasets demonstrate increased recourse rates while sacrificing accuracy mildly.


**Limitations And Societal Impact:**

None discussed sufficiently.


**Main Review:**

Originality:
The idea of a training recourse ready machine learning model is interesting. Looking for small perturbations that change the outcome has been the driving idea behind algorithmic recourse. However, thinking about a model that promotes recourse is unique.

Quality:
The paper appears to be technically sound. I have a few reservations in regards to the evaluations.

How were the features (the two features) selected for evaluation? One of the problems with providing recourse is that mutability is subjective. How will the model behave when we consider more or different features? How do the training time and recourse rates change if we increase the number of features?

We need a sensitivity analysis around $\delta$ as well. And could the authors discuss whether making a model recourse ready has any impact on diversity requirements?

I am unable to understand why the model provides CR? At best Fig 3 shows that $\lambda$ has effectively no net effect. And results show that CR works on one domain only and with limited features, which is not convincing.

Clarity:
The paper is well written and well organised.

Significance:
The idea is both interesting and relevant to the algorithmic recourse community.

**Time Spent Reviewing:**

5-6

---

> ### Author Response · Authors · 2021-08-11
> **Response to Reviewer 7Srb**
>
> Thank you for the positive review! Below, we respond to specific questions raised by the reviewer.
>
> **Re: “How were the features (the two features) selected for evaluation? One of the problems with providing recourse is that mutability is subjective.”**
>
> We selected the features in our experimental evaluation based on the conventions followed by prior works on algorithmic recourse (e.g., Ustun et. al. 2019, Karimi et. al. 2020b, Rawal et. al. 2020). Please refer to Appendix A.3 for a full list of mutable features that we employ in our experiments. For example, we consider “age” and “credit given by the bank” as actionable features for the german credit dataset and require that “age” can only increase, following the precedent of previous work (Karimi et al. al. 2020b). We will clarify these points in our main paper. We agree that mutability is subjective; in practice, we expect that domain experts would choose the permissible features for their domain.
>
> **Re: “How will the model behave when we consider more or different features? How do the training time and recourse rates change if we increase the number of features?”**
>
> Increasing the number of features should lead to higher recourse rates since it would relax constraints on what constitutes a valid recourse. We ran an experiment with the adult dataset that supports this claim: We set all continuous features to be mutable and placed no constraints on whether they can increase or decrease. We observe that the “recourse all” metric (line 253) computed with this setup results in the values: 0.673 ($\lambda=0.0$) and 0.999 ($\lambda=0.8$). Note that both these values are higher than the “recourse all” metric values from our main paper [0.461 ($\lambda=0.0$) and 0.661 ($\lambda=0.8$) -- See the second to last row, columns 1-2 in Table 1] which were obtained by placing actionability constraints on the features (i.e., employing fewer features).
>
> We observe that increasing the number of features has no noticeable effects on training time, given that we leverage fast approximations to efficiently compute $\delta$.
>
> **Re: “We need a sensitivity analysis around δ as well.”**
>
> We report results from a sensitivity analysis around $\delta$ in the “Robustness” paragraph in Section 4.2 (line 304). In the top row of Table 3, we report the percentage of recourses ($\delta$s) found that are robust for baseline models and models trained using our approach. As can be seen from Table 3, our approach does not significantly reduce the robustness of recourses: There is a slight drop in robustness for the adult dataset and for the “gradient descent” recourse algorithm for the german dataset; however, for compas and bail, our recourses remain robust.
>
> In addition, we experimented with different choices of $\delta_max$ to see whether our results are sensitive to this parameter. To this end, we plotted Figure 1 (from the main paper) for the adult dataset for different values of $\delta_max$. See https://imgur.com/aGWZCjM for results. It can be seen that our training approach successfully improves recourse rates for different values of $\delta_max$.
>
> We will include these results in the final version.
>
> **Re: “Could the authors discuss whether making a model recourse ready has any impact on diversity requirements?”**
>
> We have included results on the effect of our training on recourse disparities (i.e., differences in the rates at which recourses are offered to different subgroups) in Appendix B.1. In particular, we find that our training approach does not worsen recourse disparities between whites and non-whites, and even improves recourse rates for non-whites on the adult dataset. An interesting direction for future work would be to investigate the effect of adapting our adversarial training framework to explicitly reduce recourse disparities.
>
> **Re: “I am unable to understand why the model provides CR? At best Fig 3 shows that λ has effectively no net effect. And results show that CR works on one domain only and with limited features, which is not convincing.”**
>
> Our Causal Recourse (CR) experiments are limited by the lack of existing causal graphs for different datasets. To the best of our knowledge, the German dataset is the only real world dataset with an established structural causal model designed to be used for computing recourse; in particular, it is the only one considered in recent literature on algorithmic recourse (e.g., Karimi et al., 2020b and Upadhyay et al., 2021). In line with recent literature, we focus on the German dataset for our causal recourse experiments.
>
> In response to the reviewer’s comment about Figure 3, we ran the CR experiments with more random seeds (20 instead of the original 3 presented in the paper). Thanks to these larger number of trials, we now observe clearer trends which demonstrate that our adversarial training improves CR rates (See https://imgur.com/5QL4tQA for new results with 20 seeds and error bars representing standard errors instead of standard deviations). We will include these new results in our paper.

---

### Official Review · Reviewer_teEu · 2021-07-16

**Rating:** 7
**Confidence:** 4

**Summary:**

The authors contribute to the field of algorithmic recourse by considering how to train models such that recourse options will exist with high probability with theoretical guarantees. To do so, the authors use a method which incorporates an approach from the study of adversarial training and PAC confidence sets.

**Ethical Concerns:**

No specific ethical concerns.

**Limitations And Societal Impact:**

I believe the authors are aware of societal impact of their work, and proposed it to begin with to have a positive societal impact. I did not see much discussion of limitations, however.

**Main Review:**

Strengths:
1) Interesting research direction with potential real impact on the field of algorithmic recourse.
2) Novel, as far as I can tell.
3) Clearly written and motivated.

Weaknesses:
1) Since the paper uses a reversed version of a method that was originally introduced to reduce model brittleness, I'm not sure there was enough discussion of the effect of using such an approach on model robustness for which adversarial training was first suggested to begin with. While the authors do provide some empirical evidence in Table 3 that adding Gaussian noise did not make the models catastrophically fail. However, I think further discussion and perhaps additional results with more realistic noise additions/some examples of datasets from adversarial training literature could have made this point more convincing.
2) I believe Wachter  et al. 2017, which is cited in this work, also included a discussion on the connection between adversarial training and counterfactuals/recourse. I'm not sure the authors acknowledged this enough (unless I missed it, happy to stand corrected).
3) Presentation and experimental section can be a bit improved, see suggestions below.


Minor comments:

1) Line 55-6 vs 57-8 -- don't they say the same thing in different words? Could you not drop one of them?
2) Line 63 -- it becomes clear later in the experimental section, but it might be helpful to readers to understand what you mean by "robust recourses". Robust how?
3) Section 3 might benefit from a summary of the 3 steps of the method in an algorithm(s) block(s).
4) Line ~153 -- might be good to quickly state $\Delta$ is constructed based on use case and domain knowledge/user specified?.
5) Line 175 -- "$g_\theta$ is approximately linear as a function of x" -- is this supposedly falsifiable from data? Is this a strong assumption in certain cases? Might help to discuss a bit more.
6) Line 197 -- "uses the PAC confidence... constructs the new...", should "constructs" be changed to "to"? Typo?
7) Eq. 4 -- extra closing bracket?
8) Table 1 -- Consider adding titles to the two bottom lines, it's a bit confusing as it stands.
9) Line 215, 222 -- cite the adult income dataset? and bail outcomes?
10) Fig 2. -- why are results from the German dataset not included for this analysis? + I might have missed this, but why would recourse neg be 0 *for any threshold* under $\lambda=0$? And for all datasets? Is it just a matter of scale that makes it hard to see? I don't think based on other papers it make sense for recourse rates to be so bad?
11) Lines 227-231 -- Is this just describing 3 fold cross-validation? If no, perhaps clarify the difference + what is the difference between the procedure for bail vs. adults/compas in this aspect?
12) Line 254-5 -- why would this be the most useful measure of recourse? Why would we want to provide recourse to individuals who already received a positive outcome?
13) Table 2 -- why does it seem to be organized as a transpose of table 1 instead of consistent with it?
14) Fig. 3 -- can you describe why recourse neg shouldn't be more monotonic in $\lambda$? Do you have an intuition?
15) Use of the term "significantly" could be more cautious.
16) Footnote 4 -- "maximize 'recourse neg" on the train set" -- can you explain why train and not validation?
17) Bibliography -- Line 370, 372 -- repeated citation?


**Time Spent Reviewing:**

10

---

> ### Author Response · Authors · 2021-08-11
> **Response to Reviewer teEu**
>
> We appreciate the detailed comments and suggestions for improving our paper, and will incorporate them into our final version. Below, we respond to specific questions raised by the reviewer.
>
> **Re: “Since the paper uses a reversed version of a method that was originally introduced to reduce model brittleness, I'm not sure there was enough discussion of the effect of using such an approach on model robustness for which adversarial training was first suggested to begin with. While the authors do provide some empirical evidence in Table 3 that adding Gaussian noise did not make the models catastrophically fail. However, I think further discussion and perhaps additional results with more realistic noise additions/some examples of datasets from adversarial training literature could have made this point more convincing.”**
>
> In addition to leveraging Gaussian noise to assess the brittleness of our classifiers, we also experimented with other kinds of perturbations. More specifically, we experimented with:
>
> (a) Natural Adversarial Examples (NAE) approach (Zhao et. al. 2018) which employs GANs to generate adversarial examples that lie on the data manifold,
>
> (b) A variant of the above approach that also leverages GANs to generate small random perturbations that lie on the data manifold. The main difference w.r.t. the above approach is that here we do not explicitly optimize for the perturbations to have different labels than the original instances, and
>
> (c) the Deepfool approach (Moosavi-Dezfooli et. al., 2016), which is an iterative gradient-based approach to generate adversarial examples for a given input sample.
>
> Our results indicate that the classifiers produced by our framework are very comparable in terms of their robustness to vanilla classifiers produced without our recourse loss (i.e., $\lambda = 0$) even with the aforementioned perturbation techniques. We evaluated the robustness of our classifiers ($\lambda = 0.8$) and vanilla classifiers ($\lambda = 0$) using the same metrics computed in the last row of Table 3 and found that the difference in robustness across all datasets <= 0.03.
>
> We will include these results in the final version of our writeup.
>
> **Re: “I believe Wachter et al. 2017, which is cited in this work, also included a discussion on the connection between adversarial training and counterfactuals/recourse. I'm not sure the authors acknowledged this enough (unless I missed it, happy to stand corrected).”**
>
> While Wachter et al. (2017) briefly discuss the relationship between adversarial examples and counterfactuals/recourse, they do not touch upon adversarial training or how it can be leveraged to construct models that ensure high recourse rates. We will include a detailed discussion about this in our final version.
>
> **Re: “Presentation and experimental section can be a bit improved, see suggestions below.”**
>
> We appreciate the reviewer’s detailed feedback and will update the paper to reflect these changes and clarifications.
>
> **Re: “Fig. 3 -- can you describe why recourse neg shouldn't be more monotonic in \lambda? Do you have an intuition?”**
>
> In response to the reviewer’s comment about the lack of monotonicity in Figure 3, we ran the CR experiments with more random seeds (20 instead of the original 3 presented in the paper). We find that leveraging more trials produces a more monotonic plot, in line with the reviewer’s intuition. (See https://imgur.com/5QL4tQA for new results with 20 seeds and error bars representing standard errors instead of standard deviations.) We will include these new results in our paper.
>
> **Re: I did not see much discussion of limitations, however.**
>
> We will include a discussion of limitations in the final version.
>
>
> We hope we addressed all your questions/concerns/comments adequately. In light of our clarifications, please consider increasing your score to accept. Please let us know if we can provide any further details and/or clarifications.

---

> > ### Comment · Reviewer_teEu · 2021-09-01
> > **Thank you for a detailed and thorough response**
> >
> > Reading through the reviews and author responses, I tend to find the answers provided convincing, and in particular appreciate your efforts in providing additional experimental results and helpful discussions. I hope those provided to me, and the other reviewers, will find their way to the next version of the manuscript. As a result, I increased my score to 7.

---

> > > ### Author Response · Authors · 2021-09-07
> > > **Thank you!**
> > >
> > > We are very glad to know that the reviewer found our response helpful.
> > >
> > > Thank you so much for increasing your score to 7 (accept). We really appreciate it.

---

### Decision · Program_Chairs · 2021-09-27

**Decision:**

Accept (Poster)

**Comment:**

The paper studies how to train models that offer good opportunity for recourse, quantifying what that might mean in terms of probabilities and providing algorithms with guarantees on these. The paper is an interesting and valuable contribution to the area of algorithmic recourse and was well reviewed by the referees. Many useful and detailed suggestions were made and the authors should incorporate these into a camera ready version.